# LARGE KERNEL NETWORK FOR IMAGE RESTORATION

## ABSTRACT

The pursuit of large receptive fields has shaped the evolution of computer vision frameworks, spanning from convolutional neural networks (CNNs) to Transformers and Mamba. Recently, large-kernel operations have revitalized CNNs, making them competitive once again and widely applicable across diverse vision tasks. However, scaling kernel sizes inevitably results in substantial growth in both parameters and computational overhead. Consequently, existing approaches are often limited to small kernels or resort to decoupled designs for large kernels. In this paper, we propose a simple and efficient large kernel network for image restoration, termed ArtIR, motivated by the channel redundancy observed in image restoration models. Specifically, ArtIR applies adaptive large-kernel operations to a collapsed single channel and employs an ultra-lightweight channel attention mechanism to restore channel diversity. To complement local features, we further introduce a large kernel fusion module that integrates multi-scale information. Unlike most prior methods that focus on a narrow set of restoration tasks, we comprehensively evaluate ArtIR across single-degradation, all-in-one, and composite degradation scenarios. Beyond generic restoration, we also assess our model on domain-specific applications such as ultra-high-definition restoration, medical imaging, and remote sensing. Extensive experiments demonstrate that ArtIR achieves state-of-the-art performance while maintaining high efficiency and fast inference.

## 1 INTRODUCTION

Image restoration aims to reconstruct a high-quality image from a low-quality observation. Expanding the receptive field is essential for modeling robust long-range pixel dependencies, which has long been regarded as one of the central challenges in this field. Early approaches sought to enlarge the receptive fields of CNNs through techniques such as dilated convolutions and the use of deeper network architectures. Recently, Transformer and Mamba architectures have been introduced to this domain, owing to their strong capacity for capturing long-range dependencies (Ali et al., 2023).

More recently, large-kernel CNNs have reemerged, showing impressive performance in high-level vision tasks (Chen et al., 2024a; Li et al., 2025b; Liu et al., 2022; 2023; Ding et al., 2022a; Yasuki & Taki, 2024; Wang & Xi, 2025; Li et al., 2025d; Liu et al., 2025; Li et al., 2023b). These advances are largely motivated by insights into the success of Transformers, particularly the role of self-attention mechanisms in enabling large receptive fields. Leveraging large kernel sizes, CNN-based approaches have matched or even surpassed state-of-the-art Transformers. As a result, efficiently expanding receptive fields in CNNs has become one of the central themes in computer vision.

This trend has also extended to the field of image restoration. A widely adopted strategy is to decouple large-kernel convolutions into more computationally efficient components (Wang et al., 2024c; Luo et al., 2023; Ruan et al., 2023), such as smaller depth-wise convolutions, depth-wise dilated convolutions, and point-wise convolutions. For example, LKD (Luo et al., 2023) demonstrates that a $13 \times 13$ depth-wise convolution can be decomposed into a $5 \times 5$ depth-wise convolution and a $5 \times 5$ depth-wise dilated convolution with a dilation rate of 3. However, such decompositions fail to provide direct interactions among pixels within large windows, thereby reducing the representational capacity of the model. To address this limitation, several works apply unabridged large-kernel convolutions directly to feature maps (Hu et al., 2025b; Lee et al., 2024). For instance, OKNet (Cui et al., 2024b) employs a $63 \times 63$ depth-wise convolution to capture large-scale contextual information. Although placed in the bottleneck, its full-channel operation still introduces substantial computational overhead. Processing a 4K (*i.e.*, $3840 \times 2160$) image, for example, requires approximately 4 GFLOPs

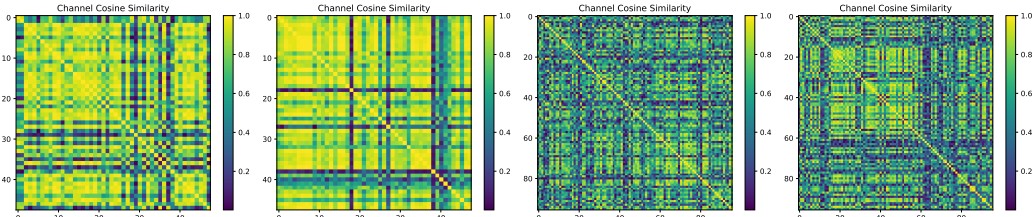

Figure 1: Visualization of channel-wise cosine similarity in Restormer (Zamir et al., 2022), where brighter colors indicate higher similarity. Additional examples are provided in the Appendix.

for each additional feature channel in the bottleneck. Another popular line of research employs frequency-domain processing to modulate global information (Zhou et al., 2023; Cui et al., 2024b; Mao et al., 2023; Li et al., 2023a). The typical pipeline involves transforming spatial features into the spectral domain, applying convolutions for modulation, and then converting the modulated spectra back to the spatial domain. However, frequency-based approaches struggle to explicitly model relationships between specific pixels. Moreover, they often introduce significant computational complexity, particularly when applied to high-resolution feature maps (Li et al., 2023a).

To address the aforementioned limitations, we propose a novel approach that leverages large-kernel operations to construct an efficient image restoration network. Our method is motivated by the observation of channel redundancy in image restoration models. As illustrated in Figure 1, we visualize the cosine similarity among channels at different scales in a representative baseline, Restormer (Zamir et al., 2022). The results reveal substantial redundancy across channels, with additional visualizations for other models provided in the Appendix. Guided by this observation, our design applies large-kernel operations to a collapsed single channel. To restore channel diversity, we introduce a lightweight channel attention mechanism that modulates feature channels, thereby striking a balance between efficiency and information preservation. This strategy enables the use of extremely large kernels while avoiding the computational burden of applying them to all channels.

Nevertheless, relying solely on large-kernel operations may lead to feature oversmoothing. To address this, we introduce a large kernel fusion module that enables interactions between local–large and large–large receptive fields, thereby enhancing multi-scale representation learning. The module partitions channels into two groups: one progressively enlarges receptive fields, while the other focuses on embedding local signals. Importantly, this design is also partly motivated by channel redundancy. Rather than relying on convolutional layers, the module matches the channel counts between groups by replicating features, which effectively reduces both parameter count and computational overhead.

On the other hand, most existing restoration models are evaluated on a narrow set of tasks, typically focusing on a single setting. To more comprehensively demonstrate the effectiveness of ArtIR, we conduct evaluations across multiple mainstream scenarios, including single-degradation, all-in-one, and composite degradation settings[1]. In addition, since image restoration has extensive applications in specialized domains, we further evaluate our method on domain-specific tasks, including ultra-high-definition (UHD), medical imaging, and remote sensing. The main contributions are as follows:

- We propose a novel solution for leveraging large-kernel operations, inspired by the channel characteristics observed in restoration models. Specifically, our design applies large-kernel operations to a collapsed single channel, complemented by lightweight channel attention to recover channel diversity, thereby achieving a better trade-off between efficiency and information preservation.

- We introduce a large kernel fusion module that enhances multi-scale representation learning by enabling interactions between local–local and local–large receptive fields.

- We conduct comprehensive experiments on both generic tasks (single-degradation, all-in-one, and composite degradation) and domain-specific tasks (UHD, medical imaging, and remote sensing). The results show that the proposed model, ArtIR, achieves state-of-the-art performance while preserving high efficiency and fast inference speed (see Figure 2 and Table 13).

---

[1]In the single-degradation setting, the model is trained separately for specific tasks such as dehazing and desnowing. In the all-in-one setting, the model is trained on a compound dataset that integrates multiple tasks, where each image is degraded by only one type of distortion. After training, the model can address all these tasks. In the composite degradation setting, each image contains multiple degradation types simultaneously.

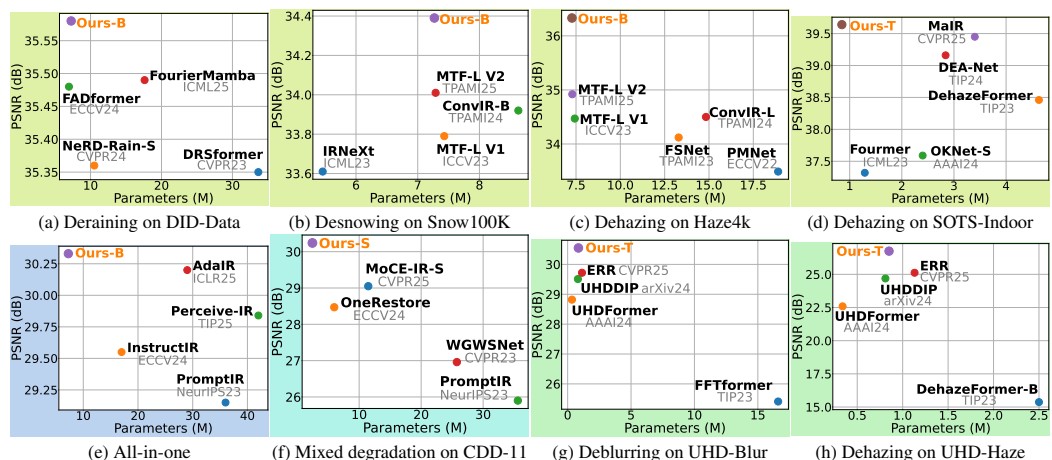

Figure 2: Comparison of parameter efficiency and PSNR under various image restoration scenarios: single-degradation , all-in-one , composite degradation , and ultra-high-definition .

## 2 RELATED WORK

**Image restoration.** Recent deep learning frameworks have substantially advanced image restoration and can be broadly categorized into CNN-based, Transformer-based, and Mamba-based approaches. A key driving factor behind these architectures is the pursuit of large-kernel operations, which enable robust large-scale receptive fields and thereby improve the removal of severe degradations.

To overcome the local connectivity limitation of convolution operators, CNN-based methods typically enlarge receptive fields by stacking deeper layers, employing strip operations, applying dilated convolutions, or adopting multi-stage paradigms (Zhang et al., 2018b; Ren et al., 2016; 2020; Cui et al., 2023c; Son et al., 2021; Hao et al., 2024). Subsequently, Transformer-based models, empowered by self-attention, have further improved performance across diverse restoration tasks such as dehazing, deraining, and deblurring (Guo et al., 2022; Chen et al., 2023; Song et al., 2023; Qiu et al., 2023; Jin et al., 2025; Tsai et al., 2022; Zamir et al., 2022; Wang et al., 2022; Kong et al., 2023; Liang et al., 2021; Chen et al., 2021a). However, this success comes at the cost of the quadratic complexity of self-attention with respect to input size. Although several strategies restrict the attention region or alter the computation dimension, obtaining large-scale receptive fields with high efficiency remains challenging. More recently, Mamba-based frameworks have been introduced to implicitly capture long-range dependencies by propagating contextual information through advanced scanning strategies (Li et al., 2025a; Guo et al., 2024a; Luan et al., 2025; Weng et al., 2024; Li et al., 2025c; Zou et al., 2024). In contrast to these approaches, we propose a new solution that leverages large-kernel operations to achieve efficient, effective, and explicit modeling of long-range pixel correlations.

**Large kernel network.** The success of Transformers has been attributed to several factors, including their advanced architecture (Yu et al., 2022), frequency bias (Park & Kim, 2022), and capacity to capture long-range dependencies (Vaswani et al., 2017). More recently, increasing attention has been directed toward their ability to model large receptive fields. Following this trend, CNNs have reemerged in high-level vision tasks by incorporating large kernels (Liu et al., 2022; 2023; Ding et al., 2022a; Chen et al., 2024a; Ding et al., 2022b; Xu et al., 2023; Li et al., 2025b; Ding et al., 2024).

Inspired by this development (Xie et al., 2023; Wang et al., 2024c), several image restoration methods adopt kernel decomposition, factorizing a large-kernel convolution into smaller components (Wang et al., 2024c; Ruan et al., 2023; Luo et al., 2023), such as depth-wise and dilated convolutions. However, such decomposition inevitably reduces representational capacity. Other approaches directly apply intact large-kernel convolutions in square or stripe form (Cui et al., 2024b; Hu et al., 2025b; Lee et al., 2024; 2025; Shi et al., 2024), but applying these operations across many channels leads to substantial computational overhead. An alternative direction leverages frequency-domain processing, such as the Fourier transform, to encode global information according to the convolution theorem (Zhou et al., 2023; Mao et al., 2023; Cui et al., 2023a; Li et al., 2023a). Yet, this strategy struggles to explicitly model pixel-wise relationships and introduces additional complexity due to reliance on transformer-based tools (Li et al., 2023a).

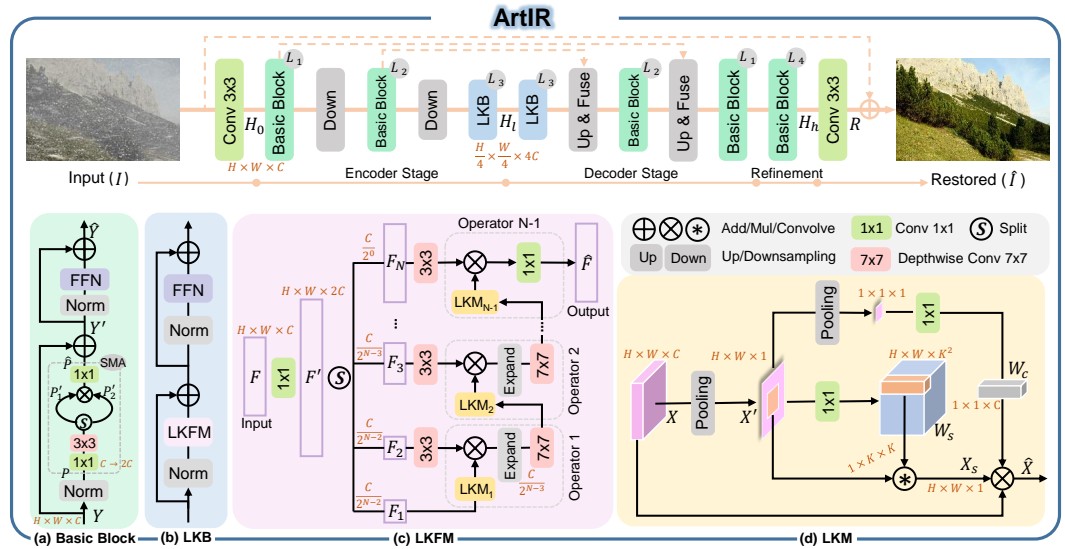

Figure 3: **Architecture of ArtIR.** The network follows a U-shaped learning paradigm and is primarily composed of (a) the Transformer-style basic block and (b) the large-kernel block (LKB). The (c) large-kernel fusion module (LKFM) enables multi-scale learning by progressively integrating large-scale contextual information of different sizes with local details. The (d) large-kernel module (LKM) applies large-kernel operations to a single-channel feature map, complemented by a lightweight channel attention mechanism to enhance channel diversity, thereby balancing efficiency and information preservation. The *Expand* operation doubles the channel dimension via duplication.

## 3 METHODOLOGY

This section presents the design of ArtIR for image restoration, including the pipeline and its two core components, the Large Kernel Module (LKM) and the Large Kernel Fusion Module (LKFM).

**Overall pipeline.** The pipeline of ArtIR is illustrated in the top part of Figure 3. ArtIR adopts a U-shaped Transformer-style architecture, consisting of an encoder, a decoder, and a refinement stage (Zamir et al., 2022). Both the encoder and decoder comprise multiple Basic Blocks, while the Large Kernel Block (LKB) is applied only in the bottleneck to reduce computational overhead (Cui et al., 2024b). Given a degraded image $I \in \mathbb{R}^{H \times W \times 3}$, ArtIR first employs a $3 \times 3$ convolution to extract shallow features $H_0 \in \mathbb{R}^{H \times W \times C}$. These features are then processed by a three-scale encoder to produce low-resolution representations $H_l \in \mathbb{R}^{\frac{H}{4} \times \frac{W}{4} \times 4C}$. The decoder takes $H_l$ as input and progressively reconstructs high-resolution features. During this process, encoder and decoder features are concatenated through residual connections and fused by a $1 \times 1$ convolution. Finally, a refinement stage (Zamir et al., 2022) further enhances the features, yielding $H_h \in \mathbb{R}^{H \times W \times C}$. The restored residual image $R \in \mathbb{R}^{H \times W \times 3}$ is then generated using a $3 \times 3$ convolution, and the final restored output is obtained by $\hat{I} = R + I$.

The architecture of the basic block is illustrated in Figure 3(a). The basic block follows a Transformer-style design and incorporates a gated mechanism, termed self-modulated attention (SMA), to control information flow, inspired by its success in computer vision (Zamir et al., 2022; Ma et al., 2024a;b). Formally, given input features $Y \in \mathbb{R}^{H \times W \times C}$, the computation in a basic block is expressed as

$$\hat{Y} = \text{FFN}(\text{Norm}(Y')) + Y', \quad \text{where} \quad Y' = \text{SMA}(\text{Norm}(Y)) + Y, \tag{1}$$

where Norm, FFN, and SMA denote layer normalization, the feed-forward network, and self-modulated attention, respectively. For simplicity, we adopt the FFN design from Restormer (Zamir et al., 2022). Given input features $P \in \mathbb{R}^{H \times W \times C}$, the SMA is further defined as

$$\hat{P} = \text{Conv}_{1 \times 1}(P_1' \otimes P_2'), \quad \text{where} \quad P_1', P_2' = \text{Split}(\text{Conv}_{3 \times 3}^{\text{DW}}(\text{Conv}_{1 \times 1}(P))) \in \mathbb{R}^{H \times W \times C}, \tag{2}$$

where $\text{Conv}_{1 \times 1}$ and $\text{Conv}_{3 \times 3}^{\text{DW}}$ denote a $1 \times 1$ convolution and a $3 \times 3$ depthwise convolution, respectively; Split indicates channel-wise feature partitioning, and $\otimes$ represents element-wise multiplication.

Figure 3(b) illustrates the LKB, which largely follows the structure of the basic block, except that the SMA is replaced with the LKFM. We next describe the LKFM and its core component, the LKM.

### 3.1 LARGE KERNEL MODULE (LKM)

**Motivation.** Existing large-kernel operations are mainly implemented through kernel decoupling, direct application, or frequency-domain processing. However, as the number of channels increases, particularly in the bottleneck, the computational overhead of such operations grows substantially. Figure 1 shows that channels exhibit strong similarity, indicating that only a subset may need to be modulated without compromising performance. Motivated by this, we apply large-kernel operations to a collapsed single channel and design a lightweight channel attention mechanism to restore channel diversity. This design provides a new perspective on balancing efficiency and representation power.

**Architecture.** As shown in Figure 3(d), given input features $X \in \mathbb{R}^{H \times W \times C}$, we first apply average pooling to obtain a collapsed single channel $X' \in \mathbb{R}^{H \times W \times 1}$, which is then processed by large-kernel operations. To enhance adaptability across different datasets and tasks, the large-kernel operation is implemented as a learnable convolution with pixel-wise adaptive parameters, formally defined as

$$X_s = W_s \circledast X', \quad \text{where} \qquad (3)$$

$$W_s = \text{Conv}_{1\times 1}(X') \in \mathbb{R}^{H \times W \times K^2}, \qquad (4)$$

where $W_s$ denotes the learnable parameters of the $K \times K$ large-kernel convolution, and $X_s$ is the spatially modulated single-channel feature map. The symbol $\circledast$ denotes convolution. To determine $K$, we conduct preliminary experiments with kernel sizes ranging from 3 to 63, with implementation details provided in the Appendix. Figure 4 shows that performance consistently improves with larger kernels. Remarkably, this increase in kernel size introduces only 0.02M additional parameters and 0.06G FLOPs. Based on these results, we set $K = 63$.

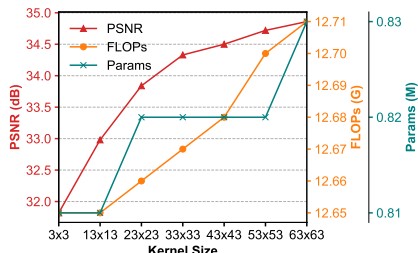

Figure 4: Experimental results with kernel sizes ranging from 3 to 63.

As large-kernel learning is performed only on a single channel, we introduce a lightweight channel attention to restore channel diversity. Specifically, average pooling is used to $X'$ to obtain single-channel global information, which is then passed through a $1 \times 1$ convolution to generate channel-wise attention weights $W_c \in \mathbb{R}^{1 \times 1 \times C}$. These weights are applied to the input $X$ for channel modulation. Compared with regular full-size attention (Chen et al., 2022), which learns channel weights directly from the input of size $H \times W \times C$, our lightweight design significantly reduces parameter count while preserving performance. Finally, the output of the LKM is given by $\hat{X} = \text{LKM}(X) = X_s \otimes X \otimes W_c$.

### 3.2 LARGE KERNEL FUSION MODULE (LKFM)

Our LKM effectively captures explicit long-range dependencies through large-kernel operations; however, it may overlook local and multi-scale information, which is critical for handling degradations of varying sizes in image restoration. To address this limitation, we propose the LKFM, which facilitates interactions between local–large and large–large receptive fields. Specifically, the input features are divided into two groups along the channel dimension: one group progressively enlarges the receptive field up to $63 \times 63$, while the other continually embeds local information into the first group. We employ an octave-like channel partition and kernel-scaling strategy to allocate larger kernels to more channels. Moreover, inspired by channel redundancy, we match the channel counts between groups through simple feature replication, thereby further improving efficiency.

Formally, given input features $F \in \mathbb{R}^{H \times W \times C}$, LKFM first expands the channel dimension using a $1 \times 1$ convolution, producing $F' \in \mathbb{R}^{H \times W \times 2C}$ for subsequent operations. $F'$ is then partitioned into $N$ channel segments, $\{F_1, \ldots, F_N\}$, which are further grouped into $F_1 \in \mathbb{R}^{H \times W \times \frac{C}{2^{N-2}}}$ and $\{F_2, \ldots, F_n, \ldots, F_N\}$ with $F_n \in \mathbb{R}^{H \times W \times \frac{C}{2^{N-n}}}$. The segment $F_1$ is recursively processed by a sequence of operators (see Figure 3(c)), primarily involving LKM with a specific kernel size $K$. For the $i^{\text{th}}$ operator ($i \in [1, N-2]$), the computation is formally defined as

$$\hat{F}_i = \text{Conv}_{7\times 7}^{\text{DW}}(\text{Expand}(\tilde{F}_i)), \quad \text{where} \quad \tilde{F}_i = \text{LKM}_i(Z) \otimes \text{Conv}_{3\times 3}^{\text{DW}}(F_{i+1}), \qquad (5)$$

Table 1: Desnowing results on Snow100K and CSD.

| Method | Snow100K PSNR | SSIM | CSD PSNR | SSIM | Params (M) | FLOPs (G) |
|---|---|---|---|---|---|---|
| IRNeXt (Cui et al., 2023c) | 33.61 | 0.95 | 37.29 | 0.99 | 5.46 | 42.1 |
| MTF-L V1 (Qiu et al., 2023) | 33.79 | 0.95 | - | - | 7.43 | 88.1 |
| ConvIR-B (Cui et al., 2024a) | 33.92 | 0.96 | 39.10 | 0.99 | 8.63 | 71.2 |
| MTF-L V2 (Jin et al., 2025) | 34.01 | 0.96 | - | - | 7.29 | 86.0 |
| Ours-B | 34.39 | 0.96 | 39.43 | 0.99 | 7.27 | 63.14 |

Table 2: Dehazing results on Haze4k.

| Method | PSNR | SSIM | Params | FLOPs |
|---|---|---|---|---|
| PMNet (Ye et al., 2022) | 33.49 | 0.98 | 18.90 | 81.1 |
| FSNet (Cui et al., 2023b) | 34.12 | 0.99 | 13.29 | 110.5 |
| MTF-L V1 (Qiu et al., 2023) | 34.47 | 0.99 | 7.43 | 88.1 |
| ConvIR-L (Cui et al., 2024a) | 34.50 | 0.99 | 14.83 | 129.9 |
| MTF-L V2 (Jin et al., 2025) | 34.92 | 0.99 | 7.29 | 86.0 |
| Ours-B | 36.33 | 0.99 | 7.27 | 63.14 |

Table 3: Dehazing results on SOTS-Indoor.

| Method | PSNR | SSIM | Params | FLOPs |
|---|---|---|---|---|
| DeHamer (Guo et al., 2022) | 36.63 | 0.988 | 132.45 | 48.93 |
| Fourmer (Zhou et al., 2023) | 37.32 | 0.990 | 1.29 | 20.6 |
| DehazeFormer (Song et al., 2023) | 38.46 | 0.994 | 4.63 | 48.64 |
| OKNet-S (Cui et al., 2024b) | 37.59 | 0.994 | 2.40 | 17.86 |
| DEA-Net (Chen et al., 2024c) | 39.16 | 0.992 | 2.84 | 24.88 |
| MaIR (Li et al., 2025a) | 39.45 | 0.997 | 3.40 | 24.03 |
| Ours-T | 39.64 | 0.997 | 0.85 | 12.76 |

Table 4: Deraining results on DID and SPA.

| Method | DID-Data PSNR | SSIM | SPA-Data PSNR | SSIM | Params |
|---|---|---|---|---|---|
| Restormer (Zamir et al., 2022) | 35.29 | 0.9641 | 47.98 | 0.9921 | 26.13 |
| DRSformer (Chen et al., 2023) | 35.35 | 0.9646 | 48.54 | 0.9924 | 33.65 |
| NeRD-Rain-S (Chen et al., 2024b) | 35.36 | 0.9647 | 48.90 | 0.9936 | 10.53 |
| FADformer (Gao et al., 2024) | 35.48 | 0.9657 | 49.21 | 0.9934 | 6.96 |
| FourierMamba (Li et al., 2025c) | 35.49 | 0.9659 | 49.18 | 0.9931 | 17.62 |
| Ours-B | 35.58 | 0.9664 | 49.54 | 0.9939 | 7.27 |

where $\hat{F}_i$ represents the output and $Z$ the input feature, which can be either $F_1$ ($i = 1$) or $\hat{F}_{i-1}$. The Expand operation adjusts channel dimensions across scales by duplicating features twice along the channel dimension, motivated by observations of channel redundancy. A $7 \times 7$ depthwise convolution is then applied to refine the coarsely expanded features. Compared with convolution-based expansion, this strategy substantially reduces parameters while achieving comparable performance. The final operator employs a $1 \times 1$ convolution to generate the output of the LKFM. In our implementation, the last operator uses a kernel size of 63, while preceding operators adopt progressively smaller kernels in an octave-like manner, *i.e.*, $\lfloor \frac{63}{2^{N-1-i}} \rfloor$. Notably, the LKFM introduces only negligible computational overhead compared to its single-scale counterpart, yet yields significant performance gains.

## 4 EXPERIMENTAL RESULTS

To evaluate the effectiveness of the proposed ArtIR, we conduct extensive experiments on three representative image restoration tasks: **(a)** single-degradation, **(b)** all-in-one, and **(c)** composite degradation. We further extend the evaluation to domain-specific tasks, including **(d)** UHD (3840×2160), **(e)** medical imaging, and **(f)** remote sensing. In the result tables, the best and second-best performances are highlighted in magenta and blue, respectively. To balance performance and efficiency, we scale our model by adjusting the number of blocks and channels across stages, yielding three variants. Additional details on the datasets and training configurations are provided in the Appendix.

### 4.1 SINGLE-DEGRADATION IMAGE RESTORATION RESULTS

In this setting, ArtIR is evaluated on six synthetic and real-world datasets spanning three image restoration tasks: desnowing, dehazing, and deraining. Separate models are trained for each dataset.

**Image desnowing.** We evaluate ArtIR on two widely used desnowing datasets: Snow100K (Liu et al., 2018) and CSD (Chen et al., 2021b). Table 1 shows that our approach significantly outperforms recent competitive methods. Notably, it achieves a 0.38 dB PSNR improvement over the Transformer-based MTF-L V2 (Jin et al., 2025), while using similar parameters and incurring lower computational cost.

**Image dehazing.** We evaluate dehazing performance on Haze4K (Liu et al., 2021), with results reported in Table 2. Our model achieves a substantial improvement over MTF-L V2 (Jin et al., 2025). We further compare against lightweight dehazing models on SOTS-Indoor (Li et al., 2018). Table 3 shows that our tiny variant surpasses the Mamba-based MaIR (Li et al., 2025a) by 0.19 dB in PSNR, while reducing parameters by 75% and FLOPs by 47%, highlighting the efficiency of our design.

**Image deraining.** We compare our model with state-of-the-art deraining algorithms on the synthetic DID-Data (Zhang & Patel, 2018) and real-world SPA-Data (Wang et al., 2019) datasets. As shown in Table 4, our model performs strongly in both synthetic and real-world scenarios. On the real-world dataset in particular, it achieves substantial gains over the frequency-based FourierMamba (Li et al., 2025c) in both PSNR and SSIM, while requiring only 41% of the parameters.

Table 5: Quantitative comparisons under the all-in-one image restoration setting.

| Method | Params | Dehazing SOTS | | Deraining Rain100L | | Denoising BSD68 | | Deblurring GoPro | | Low-Light LOLv1 | | Average | |
|---|---|---|---|---|---|---|---|---|---|---|---|---|---|
| | | PSNR | SSIM | PSNR | SSIM | PSNR | SSIM | PSNR | SSIM | PSNR | SSIM | PSNR | SSIM |
| Restormer (Zamir et al., 2022) | 26M | 24.09 | 0.927 | 34.81 | 0.962 | 31.49 | 0.884 | 27.22 | 0.829 | 20.41 | 0.806 | 27.60 | 0.881 |
| TransWeather (Valanarasu et al., 2022) | 38M | 21.32 | 0.885 | 29.43 | 0.905 | 29.00 | 0.841 | 25.12 | 0.757 | 21.21 | 0.792 | 25.22 | 0.836 |
| IDR (Zhang et al., 2023) | 15M | 25.24 | 0.943 | 35.63 | 0.965 | 31.60 | 0.887 | 27.87 | 0.846 | 21.34 | 0.826 | 28.34 | 0.893 |
| PromptIR (Potlapalli et al., 2023) | 36M | 26.54 | 0.949 | 36.37 | 0.970 | 31.47 | 0.886 | 28.71 | 0.881 | 22.68 | 0.832 | 29.15 | 0.904 |
| InstructIR-5D (Conde et al., 2024) | 17M | 27.10 | 0.956 | 36.84 | 0.973 | 31.40 | 0.873 | 29.40 | 0.886 | 23.00 | 0.836 | 29.55 | 0.908 |
| Perceive-IR (Zhang et al., 2025a) | 42M | 28.19 | 0.964 | 37.25 | 0.977 | 31.44 | 0.887 | 29.46 | 0.886 | 22.88 | 0.833 | 29.84 | 0.909 |
| AdaIR (Cui et al., 2025) | 29M | 30.53 | 0.978 | 38.02 | 0.981 | 31.35 | 0.889 | 28.12 | 0.858 | 23.00 | 0.845 | 30.20 | 0.910 |
| Ours-B | 7M | 30.62 | 0.978 | 38.08 | 0.983 | 31.47 | 0.893 | 29.38 | 0.884 | 22.12 | 0.855 | 30.33 | 0.919 |

Table 6: PSNR scores of directly applying the pre-trained all-in-one model to three denoising datasets: BSD68 (Martin et al., 2001), Urban100 (Huang et al., 2015) and Kodak24 (Rich, 1999).

| Method | BSD68 | | | Urban100 | | | Kodak24 | | | Average |
|---|---|---|---|---|---|---|---|---|---|---|
| | $\sigma = 15$ | $\sigma = 25$ | $\sigma = 50$ | $\sigma = 15$ | $\sigma = 25$ | $\sigma = 50$ | $\sigma = 15$ | $\sigma = 25$ | $\sigma = 50$ | |
| TransWeather (Valanarasu et al., 2022) | 31.16 | 29.00 | 26.08 | 29.64 | 27.97 | 26.08 | 31.67 | 29.64 | 26.74 | 28.66 |
| IDR (Zhang et al., 2023) | 34.11 | 31.60 | 28.14 | 33.82 | 31.29 | 28.07 | 34.78 | 32.42 | 29.13 | 31.48 |
| InstructIR-5D (Conde et al., 2024) | 34.00 | 31.40 | 28.15 | 33.77 | 31.40 | 28.13 | 34.70 | 32.26 | 29.16 | 31.44 |
| AdaIR (Cui et al., 2025) | 34.01 | 31.35 | 28.06 | 34.10 | 31.68 | 28.29 | 34.89 | 32.38 | 29.21 | 31.55 |
| Ours-B | 34.14 | 31.47 | 28.19 | 34.38 | 32.01 | 28.71 | 35.08 | 32.59 | 29.43 | 31.78 |

Table 7: Quantitative results on CDD-11 (Guo et al., 2024b) for composite degradation image restoration, which comprises 11 degradation categories. Results are reported in PSNR and SSIM .

| Method | Params | Low (L) | | Haze (H) | | Rain (R) | | Snow (S) | | L+H | | L+R | | L+S | | H+R | | H+S | | L+H+R | | L+H+S | | Average | |
|---|---|---|---|---|---|---|---|---|---|---|---|---|---|---|---|---|---|---|---|---|---|---|---|---|
| AirNet | 8.93M | 24.83 | .778 | 24.21 | .951 | 26.55 | .891 | 26.79 | .919 | 23.23 | .779 | 22.82 | .710 | 23.29 | .723 | 22.21 | .868 | 23.29 | .901 | 21.80 | .708 | 22.24 | .725 | 23.75 | .814 |
| PromptIR | 35.6M | 26.32 | .805 | 26.10 | .969 | 31.56 | .946 | 31.53 | .960 | 24.49 | .789 | 25.05 | .771 | 24.51 | .761 | 24.54 | .924 | 23.70 | .925 | 23.74 | .752 | 23.33 | .747 | 25.90 | .850 |
| WGWSNet | 25.76M | 24.39 | .774 | 27.90 | .982 | 33.15 | .964 | 34.43 | .973 | 24.27 | .800 | 24.60 | .765 | 27.23 | .955 | 27.65 | .960 | 23.90 | .772 | 23.97 | .771 | 26.96 | .863 | | |
| WeatherDiff | 82.96M | 23.58 | .763 | 21.99 | .904 | 24.85 | .885 | 24.80 | .888 | 21.83 | .756 | 22.69 | .730 | 22.12 | .707 | 21.25 | .868 | 21.99 | .868 | 21.23 | .716 | 21.04 | .698 | 22.49 | .799 |
| OneRestore | 5.98M | 26.48 | .826 | 32.52 | .990 | 33.40 | .964 | 34.31 | .973 | 25.79 | .822 | 25.58 | .799 | 25.19 | .789 | 29.99 | .957 | 30.21 | .964 | 24.78 | .788 | 24.90 | .791 | 28.47 | .878 |
| MoCE-IR-S | 11.47M | 27.26 | .824 | 32.66 | .990 | 34.31 | .970 | 35.91 | .980 | 26.24 | .817 | 26.25 | .800 | 26.04 | .793 | 29.93 | .964 | 30.19 | .970 | 25.41 | .789 | 25.39 | .790 | 29.05 | .881 |
| Ours-S | 2.46M | 27.45 | .837 | 35.93 | .994 | 34.97 | .974 | 36.99 | .981 | 26.79 | .836 | 26.78 | .817 | 26.54 | .809 | 32.10 | .973 | 32.93 | .976 | 26.00 | .810 | 26.13 | .809 | 30.24 | .892 |

## 4.2 ALL-IN-ONE IMAGE RESTORATION RESULTS

Following prior works (Cui et al., 2025; Zhang et al., 2025a; Conde et al., 2024), the model is trained on a mixed dataset comprising five tasks and subsequently evaluated on each task. As shown in Table 5, our approach outperforms competing methods on most metrics, achieving average gains of 0.13 dB PSNR and 0.009 SSIM over the frequency-based AdaIR (Cui et al., 2025) across all datasets. Notably, this advantage is obtained with 76% fewer parameters and without reliance on explicit degradation priors, underscoring the strong representational capacity of our large-kernel design.

To further assess generalization, we apply the pre-trained all-in-one model to two additional denoising benchmarks, Urban100 (Huang et al., 2015) and Kodak24 (Rich, 1999). Table 6 demonstrates that the proposed model exhibits stronger robustness, surpassing the second-best method, AdaIR (Cui et al., 2025), on both datasets under different noise levels.

## 4.3 COMPOSITE DEGRADATION IMAGE RESTORATION RESULTS

We further evaluate our model on CDD-11 (Guo et al., 2024b), a composite degradation benchmark where each image is affected by up to three degradation types. Results across all 11 categories are reported in Table 7. Our model achieves the best performance in every category and, on average, surpasses the recent dynamic MoCE-IR-S (Zamfir et al., 2025) by 1.19 dB in PSNR and 0.011 in SSIM. Notably, on the haze subset, the improvement reaches 3.27 dB in PSNR. Moreover, our model contains only 2.46M parameters, substantially fewer than prior methods. Figure 5 provides visual comparisons with leading approaches, illustrating the superior capability of our model in removing composite degradations from challenging examples.

## 4.4 DOMAIN-SPECIFIC IMAGE RESTORATION RESULTS

To verify the generality of our design, we evaluate ArtIR on domain-specific tasks, including ultra-high-definition (UHD), medical imaging, and remote sensing.

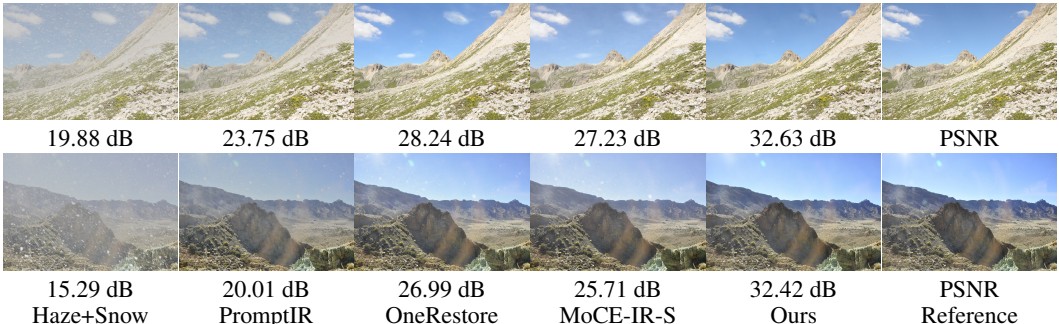

| 19.88 dB | 23.75 dB | 28.24 dB | 27.23 dB | 32.63 dB | PSNR |
| 15.29 dB | 20.01 dB | 26.99 dB | 25.71 dB | 32.42 dB | PSNR |
| Haze+Snow | PromptIR | OneRestore | MoCE-IR-S | Ours | Reference |

Figure 5: Visual results on CDD-11 (Guo et al., 2024b) for composite degradation image restoration.

Table 8: Dehazing results on UHD-Haze.

| Method | PSNR | SSIM | Params |
|---|---|---|---|
| Zheng *et al.* (Zheng et al., 2021) | 18.04 | 0.811 | 34.5M |
| Restormer (Zamir et al., 2022) | 12.72 | 0.693 | 26.1M |
| Uformer (Wang et al., 2022) | 19.83 | 0.737 | 20.6M |
| DehazeFormer-B (Song et al., 2023) | 15.37 | 0.725 | 2.5M |
| UHDFormer (Wang et al., 2024a) | 22.59 | 0.943 | 0.339M |
| UHDDIP (Wang et al., 2024b) | 24.69 | 0.952 | 0.81M |
| ERR (Zhao et al., 2025) | 25.12 | 0.950 | 1.131M |
| Ours-T | 26.75 | 0.963 | 0.85M |

Table 9: Deblurring results on UHD-Blur.

| Method | PSNR | SSIM | Params |
|---|---|---|---|
| Restormer (Zamir et al., 2022) | 25.21 | 0.693 | 26.1M |
| Uformer (Wang et al., 2022) | 25.27 | 0.737 | 20.6M |
| Stripformer (Tsai et al., 2022) | 25.05 | 0.725 | 19.7M |
| FFTformer (Kong et al., 2023) | 25.41 | 0.725 | 16.6M |
| UHDFormer (Wang et al., 2024a) | 28.82 | 0.844 | 0.339M |
| UHDDIP (Wang et al., 2024b) | 29.51 | 0.858 | 0.81M |
| ERR (Zhao et al., 2025) | 29.72 | 0.861 | 1.131M |
| Ours-T | 30.55 | 0.877 | 0.85M |

Table 10: CT image denoising results on the AAPM (McCollough et al., 2017) dataset.

| Method | PSNR↑ | SSIM↑ | RMSE↓ | Params |
|---|---|---|---|---|
| TransCT (Zhang et al., 2021) | 32.62 | 0.908 | 9.533 | 13.23 |
| Eformer (Luthra et al., 2021) | 33.35 | 0.918 | 8.803 | 0.34 |
| CTformer (Wang et al., 2023) | 33.25 | 0.913 | 8.897 | 1.45 |
| DenoMamba (Öztürk et al., 2024) | 33.53 | 0.915 | 8.612 | 112.62 |
| Restore-light (Yang et al., 2025) | 33.64 | 0.918 | 8.514 | 1.16 |
| Ours-T | 33.76 | 0.919 | 8.400 | 0.85 |

Table 11: PET image synthesis results on the PolarStar M660 (Yang et al., 2025) dataset.

| Method | PSNR↑ | SSIM↑ | RMSE↓ | Params |
|---|---|---|---|---|
| CycleWGAN (Zhou et al., 2020) | 36.62 | 0.929 | 0.091 | 1.00 |
| DCITN (Zhou et al., 2022b) | 36.09 | 0.929 | 0.097 | 0.08 |
| DRMC (Yang et al., 2023) | 36.00 | 0.935 | 0.100 | 0.62 |
| ARGAN (Luo et al., 2022) | 36.73 | 0.941 | 0.090 | 31.14 |
| Restore-light (Yang et al., 2025) | 36.96 | 0.943 | 0.089 | 1.16 |
| Ours-T | 37.25 | 0.947 | 0.086 | 0.85 |

Table 12: Remote sensing image dehazing results on SateHaze1k.

| Method | Thin | | Moderate | | Thick | |
|---|---|---|---|---|---|---|
| | PSNR | SSIM | PSNR | SSIM | PSNR | SSIM |
| AIDTransformer (Kulkarni & Murala, 2023) | 21.09 | 0.884 | 23.56 | 0.929 | 19.18 | 0.804 |
| DehazeFormer (Song et al., 2023) | 24.26 | 0.909 | 25.69 | 0.938 | 22.26 | 0.835 |
| EMPF (Wen et al., 2023) | 22.69 | 0.896 | 25.17 | 0.932 | 20.23 | 0.822 |
| Trinity (Chi et al., 2023) | 22.65 | 0.896 | 24.73 | 0.934 | 20.57 | 0.824 |
| FocalNet (Cui et al., 2023a) | 24.16 | 0.916 | 25.99 | 0.947 | 21.69 | 0.847 |
| FMambaIR (Luan et al., 2025) | 24.58 | 0.912 | 25.83 | 0.939 | 22.65 | 0.850 |
| Ours-S | 25.18 | 0.927 | 27.12 | 0.938 | 22.93 | 0.860 |

Table 13: Runtime efficiency.

| Task/Data | Method | Time/s | Speedup |
|---|---|---|---|
| Desnowing Snow100K | MTF-L V2 | 1.01 | |
| | Ours-B | 0.12 | ×8.4 |
| Dehazing Haze4k | MTF-L V2 | 0.83 | |
| | Ours-B | 0.11 | ×7.5 |
| Mixed CDD-11 | MoCE-IR-S | 0.49 | |
| | Ours-S | 0.20 | ×2.5 |
| All-in-one Rain100L | AdaIR | 0.16 | |
| | Ours-B | 0.06 | ×2.7 |

**UHD image restoration.** We evaluate ArtIR on UHD-Haze (Wang et al., 2024a) and UHD-Blur (Wang et al., 2024a) for UHD dehazing and deblurring, respectively. The results are reported in Table 8 and Table 9. Although not specifically designed for UHD tasks, our model outperforms the recent ERR (Zhao et al., 2025) algorithm on both benchmarks. In particular, ArtIR achieves PSNR gains of 1.63 dB on UHD-Haze and 0.83 dB on UHD-Blur, while requiring fewer parameters.

**Medical image restoration.** We evaluate our model on two medical imaging tasks, namely CT image denoising and PET image synthesis, using the AAPM (McCollough et al., 2017) and PolarStar M660 (Yang et al., 2025) datasets, respectively. Following (Yang et al., 2025), we compare against previous methods using PSNR, SSIM, and RMSE. As shown in Tables 10 and 11, our model consistently outperforms the specialized Restore-light (Yang et al., 2025) algorithm on both datasets while requiring fewer parameters, highlighting its potential for medical image restoration.

**Remote sensing image restoration.** For this task, we train separate models on three subsets of the SateHaze1k dataset (Huang et al., 2020). Results for thin, moderate, and thick haze levels are reported in Table 12. Our method outperforms both task-specific and general image restoration algorithms. In

Table 14: Ablation results. Further ablation studies can be found in the Appendix.

(a) Ablation study for the proposed module.

| Method | PSNR | FLOPs | Params |
|---|---|---|---|
| Conv Block (Base) | 31.38 | 13.32 | 0.86 |
| SMA | 31.80 | 12.65 | 0.81 |
| LKFM w/o channel | 36.18 | 12.81 | 0.85 |
| Full | 36.92 | 12.81 | 0.85 |

(b) Number of segments in LKFM.

| N | PSNR | FLOPs | Params |
|---|---|---|---|
| 2 | 34.86 | 12.71 | 0.83 |
| 3 | 35.75 | 12.78 | 0.84 |
| 4 | 36.18 | 12.81 | 0.85 |
| 5 | 36.12 | 12.83 | 0.86 |

(c) Alternative large-kernel operations.

| Method | PSNR | FLOPs | Params |
|---|---|---|---|
| Depth-wise Conv | 36.14 | 16.81 | 1.83 |
| Decomposition | 35.70 | 13.04 | 0.91 |
| Frequency-based | 35.51 | 12.92 | 0.94 |
| Ours | 36.92 | 12.81 | 0.85 |

particular, it surpasses the recent Mamba-based FMambaIR (Luan et al., 2025) across all three subsets in terms of PSNR. These findings indicate the robustness of our model under diverse conditions.

## 4.5 RUNTIME COMPARISON

We evaluate the runtime efficiency of our model against recent representative algorithms across multiple scenarios. As shown in Table 13, our model is approximately $8\times$ faster than the Transformer-based MTF-L V2 (Jin et al., 2025) on single-degradation tasks. For the composite degradation task, it achieves a $2.5\times$ speedup over MoCE-IR-S (Zamfir et al., 2025), which is specifically designed to accelerate inference using Mixtures of Experts. In addition, our model outperforms the all-in-one algorithm, AdaIR (Cui et al., 2025), in runtime efficiency.

## 4.6 ABLATION STUDIES

For the ablation studies, we train the tiny model on the RESIDE-Indoor dataset (Li et al., 2018) for 100k iterations and evaluate its performance on the SOTS-Indoor dataset (Li et al., 2018). Additional ablation studies on the large-kernel design are included in the Appendix.

**Effects of LKFM.** In our model, we adopt LKFM in the bottleneck and employ SMA at other scales. As a baseline, we construct a model using convolution blocks in all stages, where the split-and-multiplication operations in SMA are replaced with a $1 \times 1$ convolution for channel reduction. This baseline, denoted as Conv Block in Table 14(a), achieves 31.38 dB in PSNR. Our SMA, which incorporates a gated mechanism to regulate information flow, improves performance by 0.42 dB while maintaining higher computational efficiency. Replacing SMA in the bottleneck with our spatial LKFM (without channel attention) yields a 4.38 dB gain with negligible computational overhead. With the addition of lightweight channel attention, the complete model achieves the best performance, surpassing the convolutional baseline by 5.54 dB and further enhancing efficiency.

**Multi-scale learning in LKFM.** We implement multi-scale representation learning in LKFM by splitting features into segments. To evaluate this design, we conduct ablation experiments with different numbers of segments, as reported in Table 14(b). Overall, performance improves with more segments, confirming the effectiveness of the multi-scale strategy. In the final model, the number of segments is set to 4, providing a better trade-off between accuracy and efficiency.

**Alternatives to large-kernel operations.** We compare our method with alternative designs by replacing the LKFM in our framework. As shown in Table 14(c), a $63 \times 63$ depth-wise convolution achieves 36.14 dB in PSNR. Decomposing this convolution into smaller components following (Wang et al., 2024c), namely, a $13 \times 13$ depth-wise convolution, a dilated $9 \times 9$ depth-wise convolution with dilation rate 7, and a point-wise convolution, improves efficiency but reduces accuracy. The frequency-based variant (Mao et al., 2023), which applies a $1 \times 1$ convolution to the concatenated Fourier real and imaginary components, attains only 35.51 dB in PSNR. In contrast, our method delivers higher accuracy with lower computational cost and fewer parameters.

## 5 CONCLUSION

This paper presents an efficient and effective network for image restoration by leveraging large-kernel operations. Motivated by the observed channel redundancy in restoration models, we apply adaptive large-kernel operators directly to single-channel feature maps. To recover channel diversity, we introduce a lightweight channel attention mechanism. In addition, to enhance multi-scale learning, we design a fusion module that progressively integrates large-scale contextual information of different sizes with local details. The resulting network attains state-of-the-art performance across three generic image restoration scenarios, namely single-degradation, all-in-one, and composite degradation, while preserving high computational efficiency and fast inference. Furthermore, ArtIR demonstrates strong robustness on domain-specific tasks, including UHD, medical imaging, and remote sensing.

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

# A  APPENDIX

## A.1  IMPLEMENTATION DETAILS

This section outlines the implementation details for various image restoration settings. To ensure fair comparison, we scale our model by adjusting the number of blocks and channels at each scale of the encoder and decoder. The specifications of the three model variants are summarized in Table 15. In practice, the *expand* operation in LKFM is implemented by interleaving features along the channel dimension, duplicating each channel to achieve expansion. FLOPs are measured on $256 \times 256 \times 3$ patches, and all experiments are conducted on NVIDIA Tesla A100 GPUs. The model used in the experiments of Figure 4 corresponds to Table 14(b) with $N = 2$. Following (Cui et al., 2024b), the kernel size is increased to 63.

For fairness, no additional training tricks are applied. The code and pre-trained models will be released publicly. The ChatGPT 5 model is used to polish writing.

Table 15: Architectural specifications of the three variants of the proposed network.

| Variant | $[L_1, L_2, L_3, L_4]$ | Num. of Channels | Params | FLOPs |
|---|---|---|---|---|
| Ours-T (*Tiny*) | [1, 1, 1, 4] | [32, 64, 128, 128, 64, 32, 32] | 0.85M | 12.76G |
| Ours-S (*Small*) | [2, 3, 4, 4] | [32, 64, 128, 128, 64, 32, 32] | 2.46M | 23.73G |
| Ours-B (*Base*) | [3, 3, 6, 4] | [48, 96, 192, 192, 96, 48, 48] | 7.27M | 63.14G |

The runtime evaluation is performed on an NVIDIA RTX 4090 GPU. Scores are reported as the average runtime over all images in the corresponding datasets. For the mixed-degradation task, we use the low-light image enhancement dataset, while for the all-in-one task, we adopt Rain100L (Yang et al., 2017).

**Single-degradation image restoration.** The model is trained separately on each dataset using the Adam optimizer (Kingma, 2014) with an initial learning rate of $1 \times 10^{-3}$, which is gradually reduced to $1 \times 10^{-7}$ via cosine annealing. Training follows prior arts (Cui et al., 2023a; Cho et al., 2021), using the dual-domain $L_1$ loss and typically running for 300k iterations (Zamir et al., 2022). For deraining datasets, consistent with previous methods (Zamir et al., 2022; Chen et al., 2023), evaluation is performed on the Y channel of the YCbCr color space.

**All-in-one image restoration.** Our dataset preparation follows previous works (Potlapalli et al., 2023; Cui et al., 2025), as summarized in Table 16. The model is trained on a compound dataset collected from five tasks: dehazing, deraining, denoising, deblurring, and low-light image enhancement. For denoising, noisy images are generated by adding Gaussian noise with levels $\sigma \in \{15, 25, 50\}$ to clean images. Training configurations largely follow prior methods (Potlapalli et al., 2023; Cui et al., 2025). Specifically, the model is trained on $128 \times 128 \times 3$ patches for 130 epochs with a batch size of 32 and an initial learning rate of $2 \times 10^{-4}$.

Table 16: Summary of datasets used in all-in-one experiments.

| Setting | Dehazing | Deraining | Denoising | Deblurring | Enhancement |
|---|---|---|---|---|---|
| Train | RESIDE | Rain100L | WED, BSD400 | GOPRO | LOL |
| Test | SOTS | Rain100L | BSD68, Urban100, Kodak24 | GOPRO | LOL |

**Composite degradation image restoration.** The basic setup for this task follows that of the single-degradation setting.

**Domain-specific image restoration tasks.** The training and dataset configurations for UHD, medical imaging, and remote sensing follow representative methods in each domain (Yang et al., 2025; Zhao et al., 2025; Luan et al., 2025), without introducing additional strategies to enhance performance.

## A.2 MORE RELATED WORK: MULTI-TASK IMAGE RESTORATION

Recently, multi-task image restoration has attracted considerable attention for its ability to address multiple degradations within a single model (Fan et al., 2019; Zhu et al., 2023; Jiang et al., 2025; 2024; Zhang et al., 2025b; Ai et al., 2024; Tian et al., 2025; Rajagopalan et al., 2025; Hu et al., 2025a). In this study, we evaluate ArtIR under two multi-task settings: all-in-one and composite degradation. Existing all-in-one methods commonly follow a two-step paradigm: first extracting degradation information from inputs, and then using this information for degradation-aware restoration. For example, AirNet (Li et al., 2022) contrastively extracts degradation cues from degraded images, while PromptIR (Potlapalli et al., 2023) embeds informative features into learnable parameters. More recently, large models and additional modalities have been introduced to derive more discriminative features from inputs (Zhang et al., 2025a; Conde et al., 2024; Luo et al., 2024). Another line of work explores dynamic learning mechanisms (*e.g.*, Mixtures of Experts) to coordinate different sub-tasks and improve efficiency (Wu et al., 2024; Zamfir et al., 2025; Dudhane et al., 2024). For composite degradations, OneRestore (Guo et al., 2024b) develops a scene descriptor-guided Transformer that incorporates both visual and textual inputs (Zhou et al., 2022a; Feijoo et al., 2025).

In contrast to these approaches, we investigate the use of large-kernel operations for multi-task image restoration. Despite not relying on explicit degradation priors, our model achieves performance competitive with state-of-the-art algorithms while maintaining high efficiency. This advantage primarily arises from its strong representational capacity and adaptive learning mechanism. We hope that our model can serve as a solid baseline for future research in this area.

## A.3 EVALUATION USING PERCEPTUAL METRICS

In addition to distortion-based metrics, we evaluate our pre-trained all-in-one model using the perceptual metric LPIPS (Zhang et al., 2018a) and compare it with the state-of-the-art all-in-one algorithm (Cui et al., 2025). As shown in Table 17, our model achieves lower LPIPS scores than the competing method across most noise levels.

Table 17: LPIPS ($\downarrow$, lower is better) comparison with the state-of-the-art all-in-one method (Cui et al., 2025) on three denoising datasets.

| Method | BSD68 | | | Urban100 | | | Kodak24 | | |
|---|---|---|---|---|---|---|---|---|---|
| | $\sigma = 15$ | $\sigma = 25$ | $\sigma = 50$ | $\sigma = 15$ | $\sigma = 25$ | $\sigma = 50$ | $\sigma = 15$ | $\sigma = 25$ | $\sigma = 50$ |
| AdaIR | 0.0634 | 0.1114 | 0.2105 | 0.0419 | 0.0660 | 0.1221 | 0.0835 | 0.1299 | 0.2259 |
| Ours-B | 0.0599 | 0.1099 | 0.2157 | 0.0388 | 0.0645 | 0.1213 | 0.0790 | 0.1273 | 0.2259 |

## A.4 MORE ABLATION STUDIES

**Channel attention.** We propose an extremely lightweight channel attention mechanism within our LKM. For comparison, we evaluate a full-size channel attention variant that generates attention weights directly from the original input features (*e.g.*, $X$ in Figure 3(d)). This variant attains 36.9 dB in PSNR, which is slightly lower (-0.2 dB) than our design, while introducing an additional 0.05M parameters. These results demonstrate the effectiveness and efficiency of our approach.

**Convolution or copy?** In our LKFM, we align the channel dimensions of different segments by simply duplicating features, motivated by the channel redundancy observed in image restoration models. To assess this choice, we replace the duplication with convolutional layers for channel adjustment, which yields 36.19 dB in PSNR, 0.1 dB higher than our design, but at the cost of increased FLOPs (+0.09G) and parameters (+0.02M). Considering the trade-off, we adopt channel replication for its simplicity and efficiency.

**Alternatives to obtaining the single-channel feature.** We explore different lightweight strategies for generating the single-channel feature in LKM. As shown in Table 18, max pooling yields 35.81 dB in PSNR. Concatenating max-pooled and average-pooled features (Woo et al., 2018) followed by a $7 \times 7$ convolution for channel reduction slightly improves performance to 35.88 dB. Directly selecting a single input channel (the last channel) achieves the same performance as max pooling. In

contrast, average pooling produces the best results, and we therefore adopt this strategy in our final model.

Table 18: Alternative strategies for obtaining the single-channel feature map.

| Method | Max pooling | Max+Avg pooling | Last channel | Ours |
|---|---|---|---|---|
| PSNR | 35.81 | 35.88 | 35.81 | 36.18 |

**Adaptive strategy.** We adopt a learnable convolutional layer in our LKM to enable adaptive processing of different inputs. To assess its effectiveness, we replace this dynamic operator with a simplified variant that applies a $63 \times 63$ depth-wise convolution to the single-channel feature to generate attention weights, which are then applied via multiplication. This alternative attains only 32.3 dB in PSNR, 3.88 dB lower than our design, despite slightly reduced computational overhead and parameters (-0.13 GLOPs, -0.03M parameters).

## A.5 DISCUSSION

The design of our large-kernel operation is inspired by the channel redundancy observed in image restoration models. To further illustrate this phenomenon, we visualize channel similarity results for additional models, including both Transformer- and convolution-based architectures. As shown in Figure 6, different channels often exhibit strong similarities and may learn overlapping representations, albeit to varying degrees. In our model, we adopt an extreme strategy by applying the large-kernel operation to a single-channel feature. A promising direction for future work is to explore which channels, and how many, should share large-kernel operations. However, under the current design, such an extension would inevitably increase computational overhead.

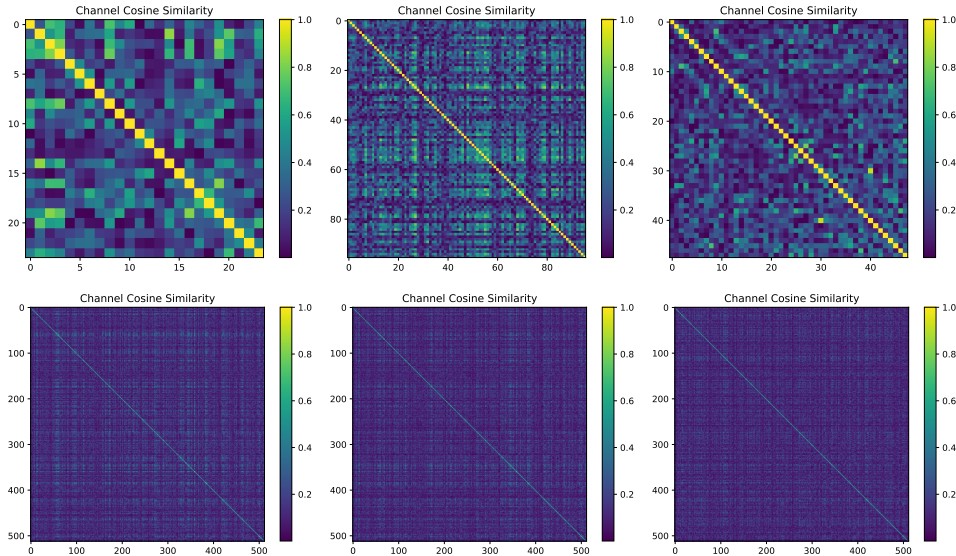

Figure 6: Visualizations of channel similarity for additional models: the Transformer-based MTF-L V2 (Jin et al., 2025) (top) and the convolutional NAFNet baseline (Chen et al., 2022) (bottom).

## A.6 MORE VISUAL RESULTS

We first present the t-SNE results of representations learned by our all-in-one model. As shown in Figure 7, the model learns discriminative features for different inputs without relying on explicit priors, highlighting the strong representational capacity of our design.

Furthermore, we visualize the feature maps learned by the local segment and the first large-kernel operator in our LKFM. As shown in Figure 8, the local channel segment captures detailed features,

while the large-kernel operation provides broader contextual perception, demonstrating the effectiveness of our design. The two groups of visualized features are extracted from different LKB in the bottleneck of the all-in-one model.

Finally, we provide additional visual comparisons across various image restoration settings, including single-degradation (Figure 9), all-in-one (Figures 10, 11, 12), and composite degradation (Figure 13).

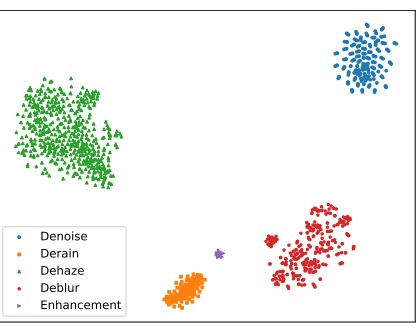

Figure 7: t-SNE visualization of the learned feature representations in our all-in-one model.

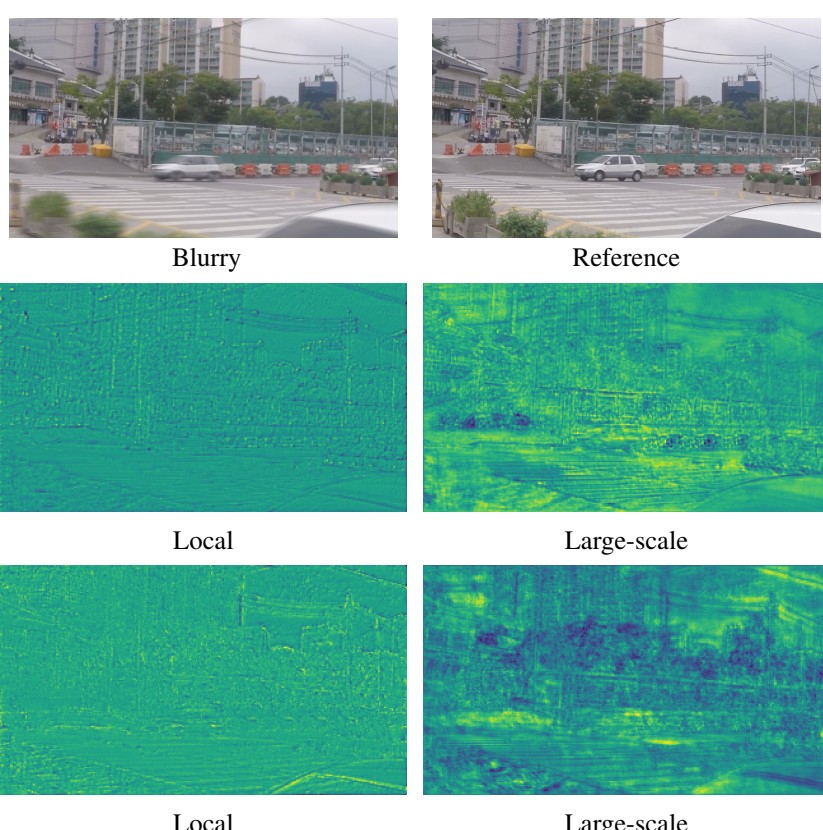

Figure 8: Visualizations of feature maps learned in our LKFM.

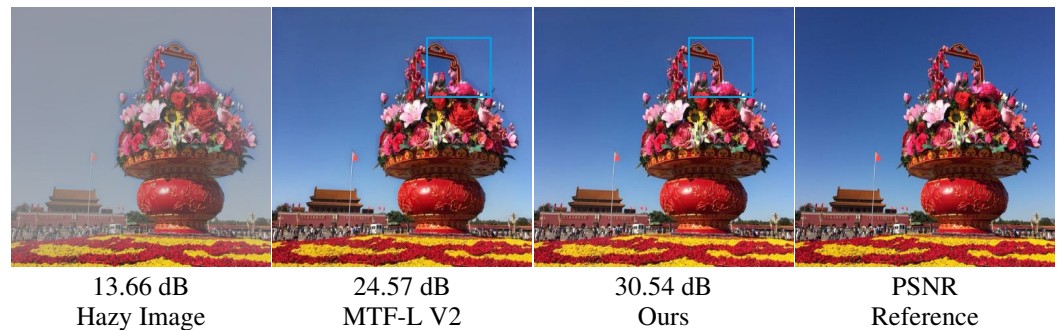

| 13.66 dB | 24.57 dB | 30.54 dB | PSNR |
| Hazy Image | MTF-L V2 | Ours | Reference |

Figure 9: Dehazing results on Haze4K (Liu et al., 2021) under the single-degradation setting.

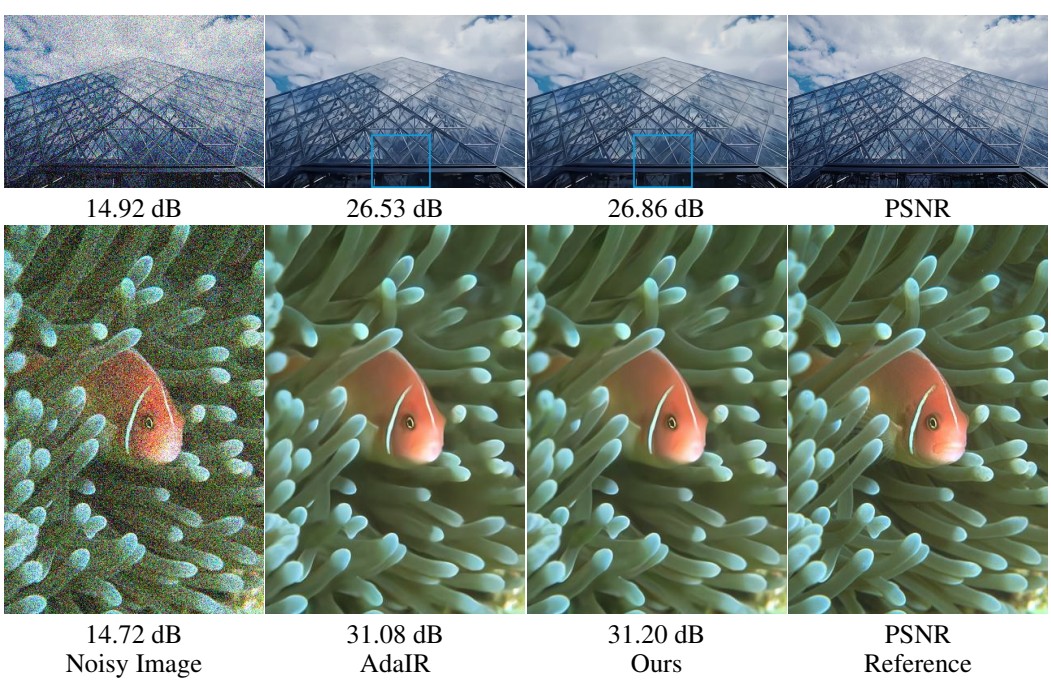

| 14.92 dB | 26.53 dB | 26.86 dB | PSNR |

| 14.72 dB | 31.08 dB | 31.20 dB | PSNR |
| Noisy Image | AdaIR | Ours | Reference |

Figure 10: Denoising comparisons on BSD68 (Martin et al., 2001) under the all-in-one setting.

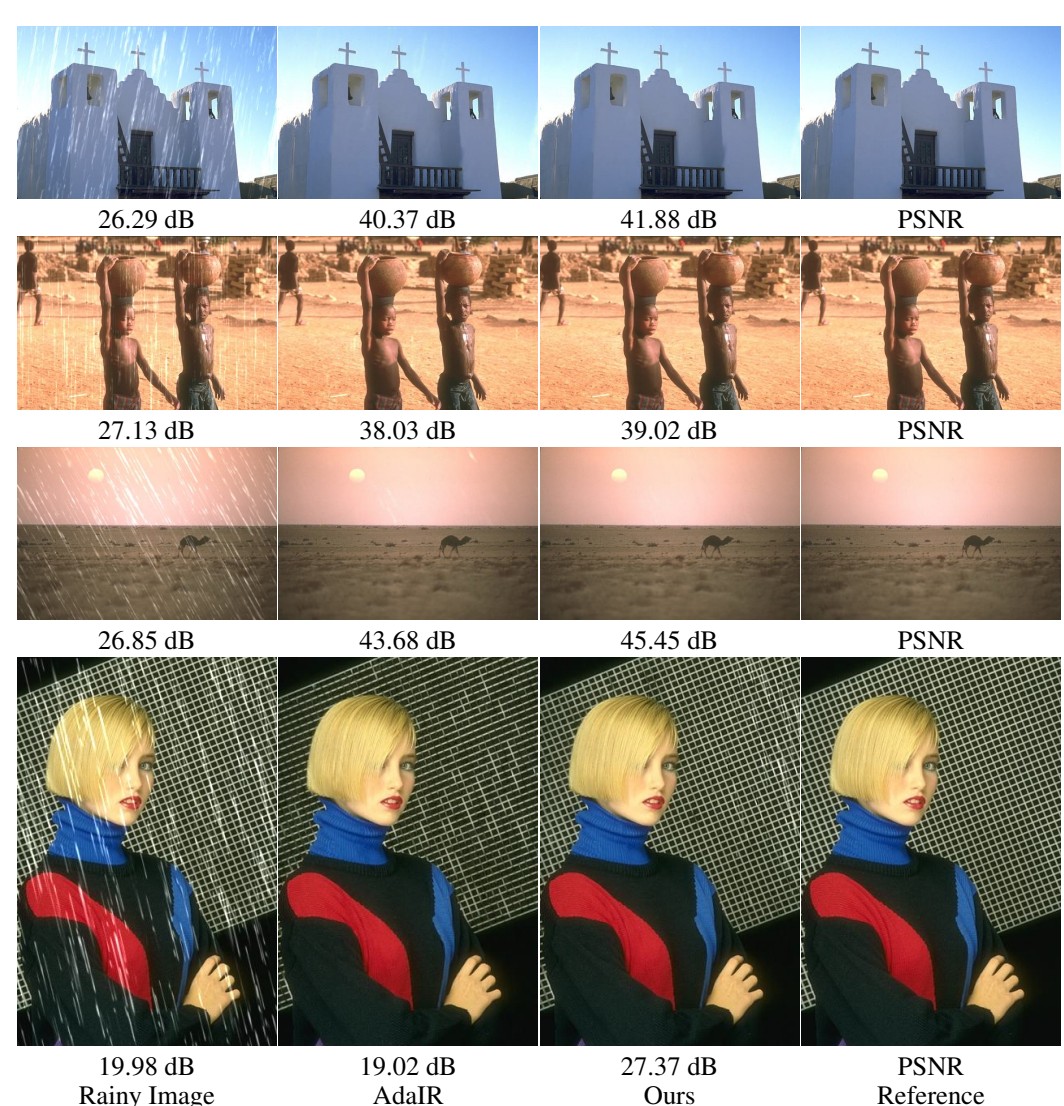

|  |  |  |  |
|---|---|---|---|
| 26.29 dB | 40.37 dB | 41.88 dB | PSNR |
| 27.13 dB | 38.03 dB | 39.02 dB | PSNR |
| 26.85 dB | 43.68 dB | 45.45 dB | PSNR |
| 19.98 dB | 19.02 dB | 27.37 dB | PSNR |
| Rainy Image | AdaIR | Ours | Reference |

Figure 11: Deraining comparisons on Rain100L (Yang et al., 2017) under the all-in-one setting.

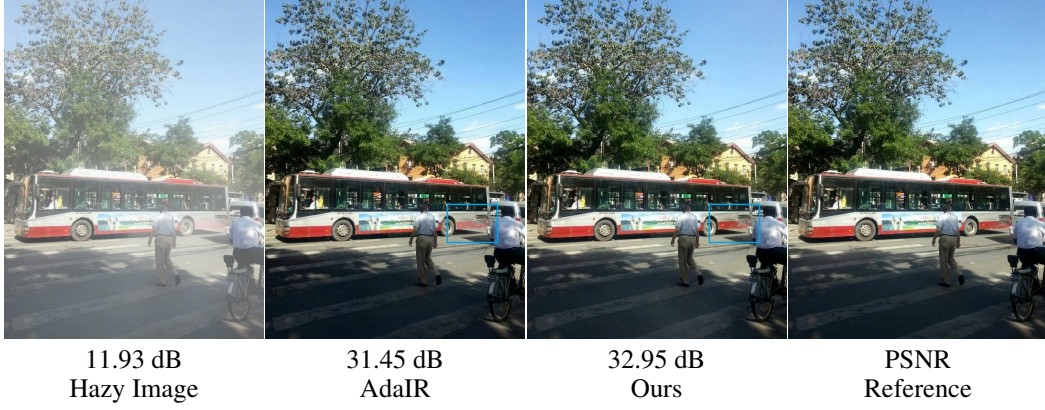

|  |  |  |  |
|---|---|---|---|
| 11.93 dB | 31.45 dB | 32.95 dB | PSNR |
| Hazy Image | AdaIR | Ours | Reference |

Figure 12: Dehazing comparisons on SOTS-Outdoor (Li et al., 2018) under the all-in-one setting.

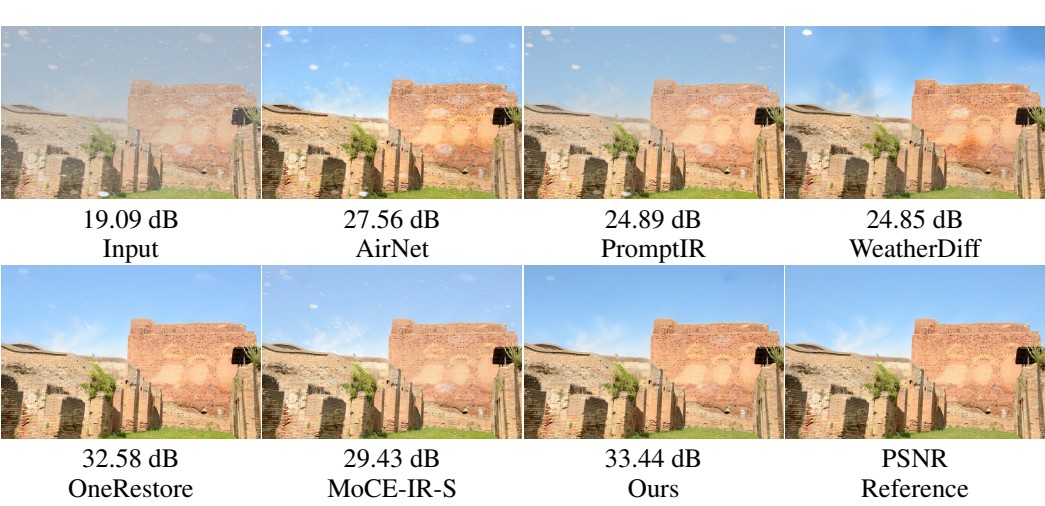

| 19.09 dB | 27.56 dB | 24.89 dB | 24.85 dB |
| Input | AirNet | PromptIR | WeatherDiff |
| 32.58 dB | 29.43 dB | 33.44 dB | PSNR |
| OneRestore | MoCE-IR-S | Ours | Reference |

Figure 13: Visual results on CDD-11 (Guo et al., 2024b) for composite degradation image restoration.

