# OpenReview forum: "Large Kernel Network for Image Restoration"
_ICLR.cc/2026/Conference — ICLR 2026 Conference Withdrawn Submission_

### Official Review · Reviewer_RUF1 · 2025-10-29

**Soundness:** 3
**Presentation:** 3
**Contribution:** 3
**Rating:** 6
**Confidence:** 5

**Summary:**

Previous image restoration methods typically rely on small convolution kernels due to the significant parameter count and computational cost associated with large kernels. To address this limitation and enable the effective use of large kernels, this paper proposes an efficient and lightweight method.

**Strengths:**

1. The experiments are comprehensive, and the method demonstrates strong performance across benchmarks.

**Weaknesses:**

1. In Fig. 1, each image should be clearly annotated to indicate the corresponding scale.
2. According to the network architecture, the large kernel is only applied at the lowest-resolution scale. What would happen if large kernels were also applied at higher scales? Would this lead to a notable increase in computational overhead?
3. In Fig. 1, the visualization suggests that higher scales (with fewer channels) exhibit more redundancy, while lower scales show less. Based on this observation, large-kernel operations would be more reasonable at higher scales where redundancy is higher. However, the proposed method applies them only at the L₃ scale, which seems inconsistent with the earlier analysis.
4. Several relevant references [1-4] should be cited to provide stronger context and better situate this work within the literature.

[1] Jiang J, Zuo Z, Wu G, et al. A survey on all-in-one image restoration: Taxonomy, evaluation and future trends[J]. IEEE Transactions on Pattern Analysis and Machine Intelligence, 2025.

[2] Zhao H, Gou Y, Li B, et al. Comprehensive and delicate: An efficient transformer for image restoration[C]//Proceedings of the IEEE/CVF conference on computer vision and pattern recognition. 2023: 14122-14132.

[3] Zou Y, Zhang H, Hu H. Big data and single‐cell sequencing in acute myeloid leukemia research[J]. MedComm–Oncology, 2023, 2(3): e47.

[4] Tian Y, Hu Q, Zhang R, et al. Combining an adenovirus encoding human endostatin and PD‐1 blockade enhanced antitumor immune activity[J]. MedComm–Oncology, 2023, 2(1): e21.

**Questions:**

See Weaknesses.

---

### Official Review · Reviewer_FMUZ · 2025-10-30

**Soundness:** 2
**Presentation:** 3
**Contribution:** 2
**Rating:** 2
**Confidence:** 4

**Summary:**

This paper introduces ArtIR, an image restoration network that utilizes large-kernel operations while keeping computational costs low. The main concept is to first apply large kernels to a collapsed single-channel representation, and then restore channel diversity through a lightweight channel attention mechanism.

**Strengths:**

1. The manuscript is well-structured and clearly presented, ensuring the content is easy to understand and follow.
2. The introduced large-kernel modules deliver performance improvements while maintaining low computational overhead.

**Weaknesses:**

1. The proposed large-kernel design is implemented only at a single scale of the network, whereas prior methods (e.g., Restormer) employ small-kernel operations across all scales. The paper should clarify the rationale behind this design choice and discuss whether applying large kernels to all stages might yield additional improvements or incur excessive computational overhead.
2. Figure 1 provides an intuitive depiction of channel redundancy; however, it lacks a quantitative analysis to substantiate the degree of redundancy or its variation across different scales and datasets.
3. Figure 4 explores the relationship between kernel size, performance, and computational cost. To strengthen the analysis, it would be beneficial to investigate whether multi-scale or adaptive kernel configurations could further enhance the balance between efficiency and performance.

**Questions:**

Please see above Weaknesses.

---

### Official Review · Reviewer_GXWv · 2025-10-30

**Soundness:** 2
**Presentation:** 3
**Contribution:** 3
**Rating:** 4
**Confidence:** 4

**Summary:**

This paper introduces ArtIR, an efficient and elegant CNN architecture for image restoration. Instead of applying large convolutional kernels directly on high-dimensional feature maps, the model first compresses all channels into a single one, performs a large-kernel convolution on this compact representation to capture long-range spatial information, and then restores channel diversity through a lightweight attention mechanism. The paper also proposes a Large Kernel Fusion Module (LKFM) to integrate multi-scale features efficiently. ArtIR is implemented in a U-Net–like framework and evaluated across various image restoration tasks, including single degradation (rain, haze, snow), all-in-one restoration, composite degradation, and domain-specific datasets such as UHD, medical, and remote sensing images. Experiments show that ArtIR consistently outperforms CNN, Transformer, and Mamba-based baselines while being significantly faster and more lightweight.

**Strengths:**

1. The proposed modules, including the Large Kernel Module (LKM) and Large Kernel Fusion Module (LKFM), are cleanly designed and easy to understand.

2. Compressing multi-channel features into a single channel before applying large-kernel convolution significantly reduces computation while maintaining a wide receptive field.

3. The model shows strong empirical performance, outperforming Transformer and Mamba-based methods across multiple restoration benchmarks.

**Weaknesses:**

1. The idea of channel compression followed by spatial processing is reminiscent of SENet, GCNet, and ECA-Net. The authors should highlight how their work differs from these methods. In addition, the design appears to combine existing ideas, large-kernel convolution, channel sequeeze-unsequeeze, and multi-scale fusion, and apply them to more tasks, which is not novel. The authors should provide a clearer explanation to demonstrate the unique contributions of this work.

2. The paper observes redundancy but does not quantify it rigorously. More analysis, such as inter-channel correlation or effective receptive field visualization, would strengthen the justification for the single-channel large-kernel design.

3. Metrics or visualizations of inter-channel correlation beyond the single Restormer example would strengthen the argument.

**Questions:**

Novelty is my main concern. Moreover, the proposed modules LKM and LKFM are described clearly but it is unclear why the multi-scale fusion or single-channel large-kernel strategy actually works. Can the authors provide more rigorous analysis to justify their effectiveness in terms of effective receptive fields or preservation of feature diversity?

---

### Official Review · Reviewer_ApP9 · 2025-11-01

**Soundness:** 1
**Presentation:** 2
**Contribution:** 2
**Rating:** 2
**Confidence:** 4

**Summary:**

This paper is about ArtIR, an efficient large kernel network for image restoration, which applies adaptive large-kernel operations to a collapsed single channel based on the motivation of channel redundancy. The authors also introduce a large kernel fusion module to integrate multi-scale information.

**Strengths:**

- I appreciate the extremely thorough evaluation. The authors tested ArtIR across a wide range of scenarios.
- The strategy of applying large kernels to a single channel significantly reduces the parameter count and FLOPs compared to recent state-of-the-art methods.

**Weaknesses:**

- Weakly Supported Core Motivation: The central premise of the paper is that "substantial redundancy across channels" (Line 076) justifies collapsing the entire feature map to a single channel for large-kernel operations. This motivation relies on visualizing cosine similarity (Figure 1). However, the evidence is selective. While Restormer shows some redundancy, the visualizations in Appendix Figure 6 for NAFNet (bottom row) show very low similarity (mostly dark colors). This contradicts the idea that high redundancy is a general characteristic of restoration models, making the fundamental design choice questionable.

- Severe Information Bottleneck in LKM: The Large Kernel Module (LKM) (Figure 3d) aggressively collapses C channels into a single channel (X') using average pooling. This discards a massive amount of information. The large-kernel operation then acts solely on this averaged map, producing a spatial modulation map (Xs). This forces all channels to share the exact same spatial processing pattern, which seems far too restrictive for complex restoration tasks that require diverse feature handling.

- Channel Diversity Restoration: The mechanism to "restore channel diversity" is confusing. The channel attention weights (Wc) are generated from the same collapsed feature (X') used for the spatial operation. If the diversity was lost during the initial averaging, it is unclear how weights derived from that average can meaningfully reintroduce it.

- The presentation of figure 3 does not help understanding. It would be better to delete the crude content.

- The adaptive kernel generation creates H X W X K^2 weights dynamically, where k = 63. This seems memory-intensive. Can you provide a more detailed breakdown of the actual memory footprint and computational cost associated with this dynamic weight generation, especially for high-resolution inputs (UHD)?

**Questions:**

- The channel similarity for NAFNet is visibly very low. How do you reconcile this observation with the paper's core motivation? Does this imply ArtIR's design principle is only applicable to architectures that inherently exhibit high redundancy?

- Did you experiment with applying large kernels to a group of channels (C' < C) rather than collapsing everything aggressively to a single channel (C'=1)? This might offer a better balance than the extreme strategy adopted.

- The ablation study shows a massive jump when introducing the spatial LKFM. This performance gain seems disproportionately large for a bottleneck modification. Could you elaborate on why this contribution is so significant?

- How does the model handle high-frequency details? Aggressive average pooling and large kernels typically lead to oversmoothing. Are fine-grained textures preserved?

---

### Note · Authors · 2025-11-14

I have read and agree with the venue's withdrawal policy on behalf of myself and my co-authors.